# Differential Expression of Key Immune Markers in the Intestinal Tract of Developing Chick Embryos

**DOI:** 10.3390/vetsci12020186

**Published:** 2025-02-19

**Authors:** Shreeya Sharma, Mohammadali Alizadeh, Scott Pratt, Alexis Stamatikos, Khaled Abdelaziz

**Affiliations:** 1Department of Animal and Veterinary Sciences, Clemson University, Clemson, SC 29634, USA; shreeys@clemson.edu (S.S.); scottp@clemson.edu (S.P.); 2Department of Pathobiology, Ontario Veterinary College, University of Guelph, Guelph, ON N1G 2W1, Canada; alizadem@uoguelph.ca; 3Department of Food, Nutrition, and Packaging Sciences, Clemson University, Clemson, SC 29634, USA; adstama@clemson.edu; 4Clemson University School of Health Research (CUSHR), Clemson, SC 29634, USA

**Keywords:** chickens, embryo, cytokine, antimicrobial peptides, intestine, toll-like receptors

## Abstract

Newly hatched chicks have weak immune systems, making them susceptible to various diseases. While much research has focused on the immune responses in the lymphoid organs of young chicks, we know little about the immune defenses in their intestines, which are important for fighting off infections. This study explored how the immune system develops in chick embryos by measuring the expression of key immune genes in the intestine at different stages of development. We discovered that certain immune-related genes were more active in specific parts of the intestine, especially as the embryo matured. This knowledge enhances our understanding of intestinal immunity during the embryonic stage and can inform the development of strategies to improve chick health by targeting early immune responses.

## 1. Introduction

Due to their undeveloped immune system, newly hatched chicks are vulnerable to various bacterial, viral, and parasitic pathogens. Most studies tend to prioritize the assessment of systemic immune responses in the lymphoid organs of chick embryos and newly hatched chicks [1,2,3], with little or no consideration given to the responses within the intestines, which is a critical site for pathogen proliferation and lesional development [4]. More specifically, these studies often lack information on the types and levels of immunomodulatory gene expression levels within the gastrointestinal tract of vaccinated embryos.

There is a consensus that defense against enteric pathogens is orchestrated by the intestinal mucosal immune system, which represents the largest immunological component in both mammals and birds [5]. The intestinal mucosal immune system consists of the intestinal epithelium, an integral component of innate immunity, and gut-associated lymphoid tissues (GALT) [6]. The GALT encompasses various structures, including the bursa of Fabricius (BF), Peyer’s patches (PP), cecal tonsils (CT), Meckel’s diverticulum (MD), intraepithelial lymphocytes (IELs), and submucosal lymphoid aggregates scattered throughout the gastrointestinal tract [7].

Indeed, intestinal epithelial cells play a dual role by serving as both physical and biochemical barriers, collectively comprising the first line of defense against a wide range of environmental pathogens, including bacteria, viruses, fungi, and parasites [8,9]. A substantial body of evidence indicates that innate responses in chicken intestinal epithelial cells and other innate immune cells are triggered by recognition of conserved microbial structures, known as pathogen-associated molecular patterns (PAMPs), by pattern-recognition receptors (PRRs) expressed on the surface or within the cytosol of these cells [10,11]. The interactions between PAMPS and their corresponding PRRs trigger intracellular signaling pathways, ultimately leading to the secretion of antimicrobial peptides (AMPs), inflammatory mediators such as cytokines and chemokines, and various other effector molecules [11].

AMPs, also known as host defense peptides (HDPs), are small cationic peptides that play a crucial role in innate immunity, especially at the early stage of avian life when other components of the immune system have not yet become fully developed [12]. It is well established that AMPs exert their antimicrobial activity by disrupting bacterial membranes, consequently leading to the death of targeted bacteria [13,14]. Two classes of AMPs have been identified in avian intestines, including avian beta-defensins (AvBDs) and cathelicidins (CATHs). The current literature indicates that chickens possess 14 AvBDs [15] and four CATHs [16]. CATHs and AvBDs are cationic peptides that exhibit broad-spectrum antimicrobial activity and are crucial for immediate protection against a wide range of bacterial, parasitic, and fungal pathogens and for modulating host immune responses. The distinct expression patterns of these peptides during embryonic development underscore their innate preparedness for pathogenic challenges post-hatching [17,18]. Therefore, early regulation of their expression ensures effective early protection during this critical stage of immune system immaturity.

Cytokines and chemokines are immunomodulating molecules that, upon activation, play a crucial role in initiating and regulating inflammatory responses, the migration of leukocytes to the site of infection or inflammation, and orchestrating adaptive immune responses [19].

While the expression of various AMPs, cytokines, and chemokines has been extensively investigated in different tissues within adult chickens [17,20,21], there is a lack of knowledge regarding their expression levels within embryonic tissues, particularly in the intestinal tract. Therefore, this study was conducted to profile key immune genes of the immune system in the intestine of developing chick embryos by measuring basal gene expression levels of crucial immune markers, including cytokines, AMPs, and Toll-like receptors (TLRs).

## 2. Materials and Methods

### 2.1. Experimental Design

Embryonated Ross 308 commercial broiler eggs were obtained from a hatchery (Fieldale Farms Corporation, Baldwin, GA, USA). After candling, they were kept in an egg incubator (GQF Manufacturing Company Inc., Savannah, GA, USA). The incubator was set to maintain a temperature of 37 °C and a humidity level of 55–73%, with automated turning functionality.

At embryonic days (EDs) 14, 17, and 20, eight eggs were sacrificed to obtain distinct sections of the intestine, including the duodenum, jejunum, ileum, and cecum. The duodenum was identified by the duodenal loop, immediately caudal to the gizzard. The jejunum was distinguished by its unique coiling pattern that commences after the duodenal loop until the location of Meckel’s diverticulum, while the ileum was identified as the region between the jejunum and the bifurcation of the ceca. The cecum, the proximal portion of the large intestine, was recognized by the presence of two blind pouches. Tissues from each section were preserved in TRIzol^TM^ reagent (Thermo Fisher Scientific, Waltham, MA, USA) immediately snap-frozen in liquid nitrogen, and stored at −80 °C until use.

### 2.2. RNA Extraction and Complementary DNA (cDNA) Synthesis

Intestinal tissue samples were homogenized using the Bead Ruptor Elite (Omni International, Kennesaw, GA, USA) for RNA extraction. Total RNA was isolated with TRIzol™ (Invitrogen, USA) following the manufacturer’s instructions and treated with DNase (DNA-free™ kit, Invitrogen, USA) to remove genomic DNA. RNA quantity and purity were evaluated using a Nanodrop One spectrophotometer (Thermo Fisher Scientific, Waltham, MA, USA).

cDNA was synthesized from purified RNA using the Superscript^®^ II First-Strand Synthesis kit (Invitrogen, USA) with oligo-dT primers (Thermo Fisher Scientific, USA), following the recommended protocol. The resulting cDNA was diluted 1:10 in nuclease-free water (Thermo Scientific, USA).

### 2.3. Quantitative Real-Time Polymerase Chain Reaction (RT-qPCR)

RT-qPCR was conducted using the LightCycler^®^ 480 system (Roche Diagnostics, Indianapolis, IN, USA) according to the protocol in [7]. The reaction mixture contained 10 µL of SYBR™ Green Master Mix (PowerTrack™; Thermo Fisher Scientific, Waltham, MA, USA), 1 µL each of forward and reverse primers (10 µM), and 3 µL of nuclease-free water. A total of 15 µL of this master mix was combined with 5 µL of cDNA in a 96-well PCR plate (USA Scientific, Ocala, FL, USA).

The qPCR cycling conditions included an initial denaturation at 95 °C for 5 min, followed by 45 cycles of denaturation at 95 °C for 10 s, annealing (temperature specific to each primer set in Table 1), and extension at 72 °C for 10 s. The mRNA expression levels of the target genes were determined relative to the housekeeping gene (β-actin) using the Roche LightCycler 480 software 1.5.0 SP4. 2 based on the 2^−ΔΔCT^ method [22].

### 2.4. Statistical Analysis

Statistical analysis was performed using GraphPad Prism V5.0 (Graphpad software, San Diego, CA, USA). The data was obtained from eight replicates for each region of the intestines at embryonic days 14, 17, and 20 and were analyzed with one-way ANOVA, followed by the Tukey’s multiple comparison test to determine the differences in means between different intestinal regions for each gene of interest within the same time point, as well as differences across the three time points within each intestinal segment. The results were considered significant if the *p* value < 0.05. Data are shown graphically as the mean of the relative gene expression data (2^−∆∆Ct^) ± the standard error of the mean (SEM).

## 3. Results

To gain a better understanding of the baseline expression of immune genes in the intestinal tract of developing chicken embryos, mRNA expression levels of representative cytokines, chemokines, AMPs, and TLRs in various segments of the small intestine (duodenum, jejunum, ileum) and cecum at EDs 14, 17, and 20 were assessed.

### 3.1. Cytokine and Chemokine Gene Expression

#### 3.1.1. Embryonic Day 14

Varying gene expression levels of the cytokines, including IFN-γ, IL-6, IL-8, IL-10, IL-13, and TGF-β, were observed across different intestinal segments and cecum. No significant differences were detected within the expression of these genes across the intestinal segments at a given time point (Figure 1a–f). Notably, IL-10 was expressed in two out of eight birds in the jejunum and ileum and five in the cecum, whereas no expression was detected in the duodenum (Figure 1e). The expression of IL-8 in the duodenum was significantly higher (*p* < 0.01) in ED14 compared to ED20 (Figure 1f).

#### 3.1.2. Embryonic Day 17

All cytokines measured in this study were constitutively expressed; however, no significant differences were observed among different intestinal segments, and the expression in the ceca was significantly less than ED20 (Figure 1a–f). Similar to ED14, not all birds exhibited expression of IL-10; it was expressed in only two out of eight birds in the ileum and cecum, four out of eight birds in the duodenum, and five out of eight birds in the jejunum (Figure 1e).

The expression of chemokine IL-8 was significantly higher (*p* < 0.05 and *p* < 0.01) in the duodenum than that in the cecum and ileum, respectively, at ED17 (Figure 1f) and in the duodenum of chick embryo at ED20.

#### 3.1.3. Embryonic Day 20

By ED 20, the cytokines and chemokine were differentially expressed in the small intestine and cecum (Figure 1a–f). Compared to different parts of the small intestine, the cecum exhibited a numerical increase in the expression levels of IFN-γ, IL-6, and IL-13 (Figure 1a–c). The expression of TGF-β in the ceca exhibited statistical significance (*p* < 0.05) in comparison to both duodenum and jejunum (Figure 1d). The expression level of IL-8 in the cecum (Figure 1f) was notably higher than that observed in the duodenum and jejunum, with significance levels of *p* < 0.01. Additionally, this expression level was significantly elevated compared to the ileum, with a significance level of *p* < 0.001.

Cytokine expression in the ceca at ED20 demonstrated elevated levels compared to earlier time points (ED14 and 17) across all measured cytokines. The expression level of the IFN-γ gene (Figure 1a) was significantly higher (*p* < 0.01 and *p* < 0.001) than that in ED14 and ED17, respectively. Similarly, IL-13 (Figure 1b) expression exhibited statistical significance (*p* < 0.001) relative to both preceding time points. IL-6 (Figure 1c) and TGF-β (Figure 1d) also displayed significantly higher expression levels (*p* < 0.05 and *p* < 0.001, respectively) when compared with ED14 and ED17.

### 3.2. Antimicrobial Peptides (AMPs) Gene Expression

#### 3.2.1. Embryonic Day 14

No significant differences were observed in the expression of AvBD-1, 2, 3, 4, and 6 among different intestinal segments (Figure 2a–h). The expression of AvBD-3 (Figure 2c) and AvBD-6 (Figure 2e) in the ileum was significantly higher (*p* < 0.05) compared to that in the ileum at ED17. Additionally, the expression of AvBD-4 in the ileum was also statistically higher (*p* < 0.05) at ED14 compared to that in the ileum at ED17 and ED20 (Figure 2d). The expression levels of CATH-1 and CATH-2 were numerically higher in the duodenum compared to the other parts of the small intestine and cecum.

#### 3.2.2. Embryonic Day 17

The expression of AvBD-1 in the duodenum was found to be significantly higher (*p* < 0.001) compared to both ED20 and ED14. Moreover, in the cecum, AvBD-1 expression exhibited a significant increase compared to ED20 (*p* < 0.001). Conversely, in the jejunum, AvBD-1 expression at ED17 was significantly lower than at ED20. Notably, no significant differences were observed between segments at the same developmental stage (Figure 2a).

AvBD-2 expression in the cecum at ED17 was significantly higher (*p* < 0.05) than that at ED20 (Figure 2b). Conversely, AvBD-3 expression in the ileum was significantly lower at ED17 than at ED17 (Figure 2c). Similarly, AvBD-4 expression in the ileum was significantly reduced (*p* < 0.05) at ED17 compared to ED14 (Figure 2d), while AVBD-6 expression in the ileum at ED17 was significantly lower than that at ED14 (Figure 2e).

In terms of CATH-3 expression, the duodenum at ED17 exhibited notably higher levels (*p* < 0.05) compared to ED20 (Figure 3c). However, no significant differences were observed in the expression of CATH-1 and CATH-2 within or across the time points (Figure 3a,b).

#### 3.2.3. Embryonic Day 20

The jejunum consistently exhibited higher expression levels of AvBDs (Figure 2a–e) and CATH (Figure 2f,g) compared to other segments of the small intestine and cecum. The expression levels of AvBD-1 and 2 were significantly higher (*p* < 0.01) in the jejunum than in both the duodenum and cecum (Figure 2a,b). AvBD-3 expression was also markedly higher in the jejunum compared to the duodenum (*p* < 0.001), ileum (*p* < 0.05), and cecum (*p* < 0.001) (Figure 2c). AvBD-4 expression was significantly higher (*p* < 0.05) in the jejunum compared to the duodenum (Figure 2d). Lastly, the expression of AvBD-6 in the jejunum was significantly elevated (*p* < 0.001) in comparison to the other intestinal segments (Figure 2e).

The expression of AvBD-1 in the jejunum exhibited a significant elevation compared to both ED14 (*p* < 0.001) and ED17 (*p* < 0.01). A noticeable shift in CATH gene expression patterns was detected in the jejunum at ED20 compared to previous time points. Specifically, for CATH-2 genes, there was a statistically significant difference (*p* < 0.01) between the jejunum and the other segments of the intestine (Figure 2g), whereas for the CATH-3 gene, a significantly higher expression was noted in the jejunum compared to the duodenum (*p* < 0.01), and the ileum (*p* < 0.05). CATH-3 expression in the jejunum was significantly higher in ED20 than in ED14 (*p* < 0.001) and ED17 (*p* < 0.01) (Figure 2h).

### 3.3. Toll-like Receptors Gene Expression

There were no significant differences in the expression levels of TLR-2, TLR-3, TLR-4, and TLR-21 among the duodenum, jejunum, ileum, and cecum (*p* > 0.05) (Figure 3a–c).

On ED 20, a notably higher expression level of TLR4 was observed in the cecum at ED20 compared to that at ED14 (*p* < 0.01) and 17 (*p* < 0.001), while TLR21 exhibited elevated expression in the cecum at ED14 (*p* < 0.01) compared to other time points.

### 3.4. Heat Map Visualization

The heatmap illustrates the spatial and temporal variability in gene expression levels of key immune-related genes, including cytokines, chemokines, AvBDs, CATHs, and TLRs, across various sections of the intestinal tract, with measurements averaged from replicates within each group to ensure accuracy.

Gene expression varied significantly across developmental stages and intestinal sections. For instance, AvBDs demonstrated increasing expression from embryonic day 14 to day 20, particularly in the ileum and cecum, whereas CATHs showed relatively stable expression across tissues. Blank cells on the heatmap indicate the absence of detectable expression for specific genes (Table 2).

## 4. Discussion

The maturation and functionality of the gut immune system during the embryonic stage are critically important in warding off microbial invasion, particularly during the chick’s early life when they are vulnerable to infections [29]. The intestinal immune system comprises epithelial cells lining the mucosal surface, intraepithelial and lamina propria lymphocytes, submucosal lymphoid aggregates, and the GALT [5]. Indeed, the hatched chick’s ability to resist infection depends on the capability of intestinal epithelial and other tissue-resident immune cells to express structurally conserved components like TLRs and to produce immune mediators, such as cytokines, chemokines, and AMPs. These elements collectively serve as the primary line of defense against invading pathogens in the intestine [29].

Although the role of the intestinal immune system in protecting adult chickens from various enteric pathogens has been extensively studied [30], the reasons behind the increased susceptibility of newly hatched chicks to enteric infections remain unclear. Previous research has linked this susceptibility to the incomplete development of the GALT [29,31]. However, there is a notable lack of knowledge regarding mucosal immune responses in this specific context. Therefore, the current study was undertaken to profile the developmental trajectory and differential constitutive expression of key immune markers, including cytokines, chemokines, AMPs, and TLRs, across the intestine during different stages of embryonic development, including ED14, 17, and 20. These time points are evenly spaced to profile the kinetic expression of some representative immune system genes, including proinflammatory cytokines (IL-6), anti-inflammatory/immunoregulatory cytokines (TGF-β and IL-10), T helper-1 (Th-1) type cytokines (IFN-γ), Th-2 type cytokines (IL-13) and chemokines (CXCL8/IL-8). Additionally, the evaluation involved measuring the expression of various innate immune genes, including TLR-2 and TLR-4, for their roles in recognizing both Gram-positive and Gram-negative bacteria, TLR-3 for its role in recognizing double-stranded RNA viruses, and TLR-21, which identifies DNA motifs in bacterial genomic DNA, along with AMPs, including AvBDs and CATHs. Profiling the expression of these immune system genes in the intestine could provide valuable insights for developing strategies to trigger the intestinal immune system of pre-hatched chicks, ultimately resulting in enhancing their resistance to enteric infections after hatching.

The results demonstrated varying expression levels of cytokines across different sections of the small intestine and cecum at both ED14 and ED17, with a consistently significant surge in expression observed in the cecum at ED20 compared to other segments of the small intestine. The significant rise in cytokine expression could be attributed to the development of the cecal tonsil (CT), a major GALT situated in the proximal part of the cecum [32]. Ontogeny studies have revealed that the development of CT initiates on embryonic ED10, and as embryonic ED13 approaches, clusters of MHC II+ cells begin to populate, gradually rising as embryonic development progresses [33]. By ED18, lymphocytes, including T and B cells, become evident. It is worth noting that this developmental pattern of CT aligns with the increased expression of cytokines and chemokines genes observed at ED20, which could provide an explanation for the potential involvement of CT in the observed differences in gene expression.

Interestingly, unlike other cytokines measured in this study, the expression of IL-10 was not detected in the duodenum at EDs 14 and 20 and was only detected in a small number of birds in the other intestinal segments, which indicates a potential inducible nature of this cytokine and/or the functional immaturity of the IL-10 producing T regulatory cells. This observation aligns with earlier reports indicating that IL-10 necessitates stimuli to trigger its expression [34]. These findings are of particular significance as they highlight the importance of inducing IL-10 expression in the intestinal tissue during enteric infections in newly hatched chicks, considering its potential to balance inflammatory responses. Furthermore, the results of the developmental trajectory analysis in developing chick embryos indicated increased expression levels of cytokines, including IFN-γ, IL-6, IL-13, and TGF-β, within the cecum at ED20 compared to earlier time points (ED14 and ED17). This suggests the evolving nature of the immune response within the intestinal tract during the developmental stages analyzed.

While the cecum exhibited increased expression of cytokines, the jejunum consistently demonstrated high expression levels of AMPs, including AvBDs (AvBD-1, AvBD-2, AvBD-3, AvBD-4, and AvBD-6) and CATHs (CATH-2, and CATH-3) at ED20. The reason for the jejunum’s capacity to exhibit elevated gene expression levels remains unclear, primarily due to the limited research on the distinct immunological features of different intestinal segments. Nonetheless, considering the broad-spectrum antimicrobial activity of AMPs against a wide range of Gram-negative and Gram-positive bacteria, fungi, parasites, and viruses [35], these data may provide some insights into the apparent resistance of hatched chicks against some enteric diseases during the first few days of their life. For instance, necrotic enteritis caused by *Clostridium perfringens* is commonly seen in the jejunum of two-week-old chicks [36]. However, validating this hypothesis requires monitoring the post-hatching expression levels of AMPs to confirm if the hatched chicks maintain high expression while considering the protective role of maternal antibodies during this phase of the chick’s life. Varying expression levels of AvBDs and CATHs were observed in the intestinal segments across the time points, with the ileum showing consistently higher expression levels of AvBD-3, -4, and -6 at ED14 compared to ED17 and 20. Given that these peptides are primarily synthesized by intestinal epithelial cells [37], the variability in gene expression along the intestinal tract may be attributed to the developmental morphogenesis of the intestinal mucosa. Nevertheless, it is noteworthy that the consistent elevation in gene expression observed in the jejunum, in contrast to other intestinal segments at ED20, suggests stability indicative of the maturation of intestinal immunological processes.

TLRs are a class of PRRs that play an important role in recognizing PAMPs and subsequent initiation of the innate immune response [38]. To date, ten TLRs have been identified in birds, including TLR-1A, TLR-1B, TLR-2A, TLR-2B, TLR-3, TLR-4, TLR-5, TLR-7, TLR-15, and TLR-21 [39]. In the present study, we explored the basal expression patterns of TLR-2 (recognizes the lipoproteins and peptidoglycan of Gram-positive bacteria), TLR3 (recognizes double-stranded RNA viruses), TLR-4 (recognizes the lipopolysaccharides of Gram-negative bacteria), and TLR-21 (recognizes bacterial and viral DNA) in different intestinal segments. Although a clear expression pattern for the measured TLRs was not identified, notable observations were made in the cecum. Specifically, a significant increase in TLR-4 expression was observed at ED20, while a significant elevation of TLR-21 expression was noted at ED14, indicating distinct temporal trends for these receptors.

Overall, while this study provides valuable insight into the developmental trajectory of crucial immune genes throughout the intestinal tract in developing chicken embryos, further studies are needed to assess the post-transcriptional level of these genes to discern whether significant increases in protein expression align with the observed gene expression results. Since no prior research has investigated the ontogeny of intestinal immunity in embryos, this study could not determine the precise onset of gene expression, and it remains possible that these genes are expressed at earlier developmental stages. Despite this limitation, our findings provide a foundational basis for future studies to elucidate the ontogeny of intestinal immunity. Furthermore, approaches like transcriptomics and metabolomics could help understand regulatory mechanisms, offering a deeper understanding of the drivers of variability in immune gene expression.

## 5. Conclusions

This study provides valuable insights into the basal level of various immune system gene expressions in the intestinal tract of chickens and unveils the diverse immune capacities of different segments of the intestine. Understanding the intestinal immune system is crucial as it lays the foundation for additional research to enhance its capabilities, particularly in segments that exhibit lower expression levels of immune system genes. Further investigations are required to evaluate gene expression in other chicken breeds, which would prove advantageous for the genetic selection of breeds with robust immune profiles. Additionally, assessing the oncogenic development of intestinal immunity at earlier time points is crucial.

## Figures and Tables

**Figure 1 vetsci-12-00186-f001:**
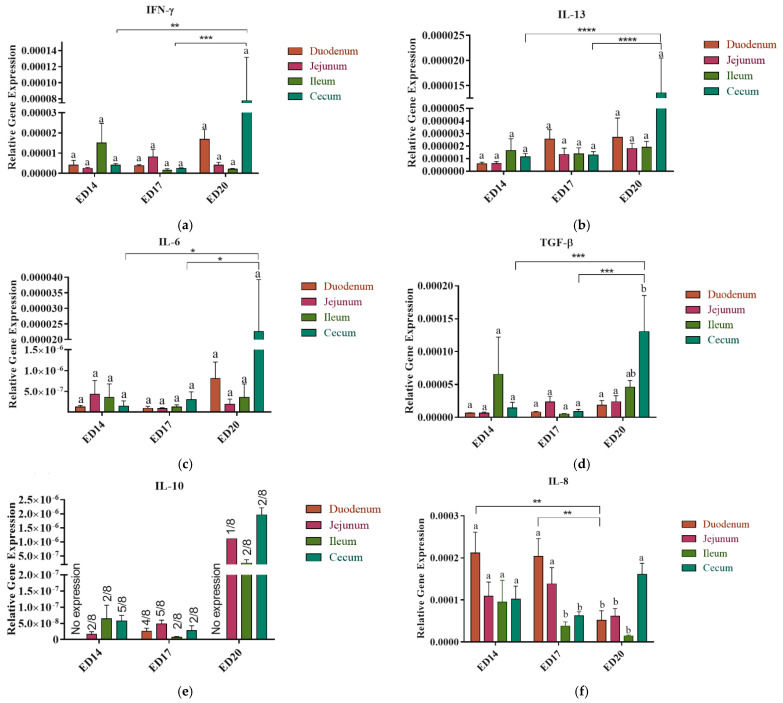
Expression patterns of cytokines and chemokine in the intestine and cecum at embryonic Day 14, 17 and 20: Tissues from the duodenum, jejunum, ileum and cecum were subjected to RNA isolation and real-time PCR was carried out to measure the basal expression levels of Th-1 associated cytokine: IFN-γ (**a**); Th-2 associated cytokine: IL-13 (**b**); proinflammatory cytokine: IL-6 (**c**); immunoregulatory cytokines: TGF-β (**d**); and IL-10 (**e**); and chemokine: IL-8 (**f**) at embryonic days 14, 17, and 20. The relative expression of target genes was normalized to the housekeeping gene (β-actin). Statistical significance among treatment groups was determined using one-way ANOVA followed by Tukey’s test. Error bars represent the standard error of the mean (SEM). Asterisks indicate significant differences: * *p* < 0.05, ** *p* < 0.01, *** *p* < 0.001 and **** *p* < 0.0001. Bars marked with the same letter are not significantly different.

**Figure 2 vetsci-12-00186-f002:**
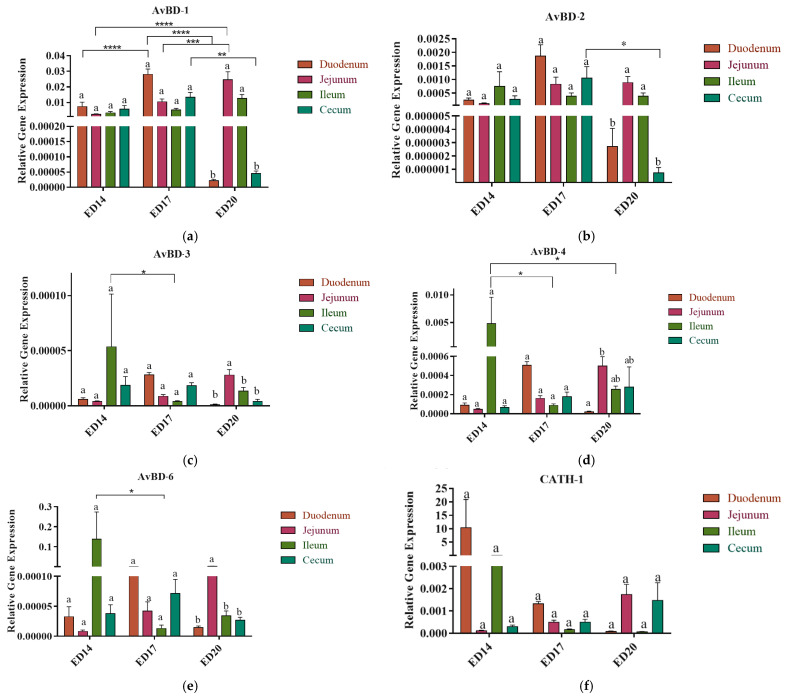
Intestinal tissue expression patterns of host defense peptides during Embryonic Day 14, 17, and 20: Tissues from the small intestine and cecum were subjected to RNA isolation, and real-time PCR was carried out for avian beta-defensins: AvBD-1 (**a**); AvBD-2 (**b**); AvBD-3 (**c**); AvBD-4 (**d**); AvBD-6 (**e**); and cathelicidins: CATH-1 (**f**); CATH-2 (**g**) and CATH-3; (**h**) for embryonic days 14, 17, and 20. The relative expression of target genes was normalized to the housekeeping gene (β-actin). Statistical significance among treatment groups was determined using one-way ANOVA followed by Tukey’s test. Error bars represent the standard error of the mean (SEM). Asterisks indicate significant differences: * *p* < 0.05, ** *p* < 0.01, *** *p* < 0.001 and **** *p* < 0.0001. Bars marked with the same letter are not significantly different.

**Figure 3 vetsci-12-00186-f003:**
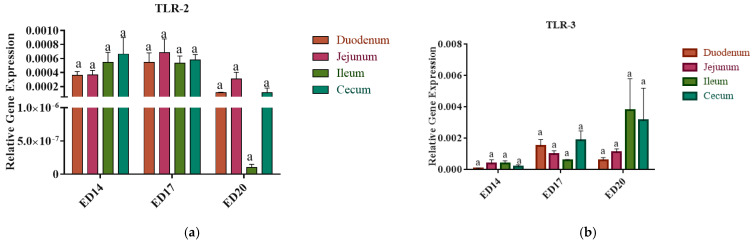
Intestinal tissue expression patterns of Toll-like receptors (TLRs) during Embryonic Day 14, 17, and 20: Tissues from the small intestine and cecum were subjected to RNA isolation, and real-time PCR was carried out for TLR-2 (**a**); TLR-3 (**b**); TLR-4 (**c**) and TLR-21 (**d**) for embryonic days 14, 17, and 20. The relative expression of target genes was normalized to the housekeeping gene (β-actin). Statistical significance among treatment groups was determined using one-way ANOVA followed by Tukey’s test. Error bars represent the standard error of the mean (SEM). Asterisks indicate significant differences: * *p* < 0.05, ** *p* < 0.01, and *** *p* < 0.001. Bars marked with the same letter are not significantly different.

**Table 1 vetsci-12-00186-t001:** Primer sequences used for real-time quantitative PCR.

Gene	Primer Sequence (5′–3′)	Annealing Temperature (°C)	Reference
β-actin	F: CAACACAGTGCTGTCTGGTGGTA R: ATCGTACTCCTGCTTGCTGATCC	60	[23]
IFN-γ	F: ACACTGACAAGTCAAAGCCGCACA R: AGTCGTTCATCGGGAGCTTGGC	60	[23]
IL-6	F: CGTGTGCGAGAACAGCATGGAGA R: TCAGGCATITCTCCTCGTCGAAGC	60	[24]
IL-8	F: CCAAGCACACCTCTCTTCCA R: GCAAGGTAGGACGCTGGTAA	64	[24]
IL-10	F: TTTGGCTGCCAGTCTGTGTC R: CTCATCCATCTTCTCGAACGTC	64	[25]
IL-13	F: ACTTGTCCAAGCTGAAGCTGTC R: TCTTGCAGTCGGTCATGTTGTC	60	[25]
TGF-β	F: CGGCCGACGATGAGTGGCTC R: CGGGGCCCATCTCACAGGGA	60	[25]
CATH-1	F: GCTGACCCTGTCCGCGTCA R: GAGGTTGTATCCTGCAATCAC	60	[26]
CATH-2	F: CAAGGAGAATGGGGTCATCAG R: CGTGGCCCCATTTATTCATTCA	60	[26]
CATH-3	F: CCATGGCTGACCCTGTCC R: TGATGGCTTTGTAGAGGTTGATG	60	[26]
AvBD-1	F: GGATGCACGCTGTTCTTGGT R: TCCGCATGGTTTACGTCTGTC	60	[27]
AvBD-2	F: CTGCTTCGGGTTCCGTTCCT R: TGCTGCTGAGGCTTTGCTGTA	60	[27]
AvBD-3	F: AGGATTCTGTCGTGTTGGGAGC R: TTCCAGGAGCGAGAAGCCAC	60	[27]
AvBD-4	F: GGCTATGCCGTCCCAAGTATT R: CCAAATCCAACAATGCAAGAAG	60	[27]
AvBD-6	F: TGGCAGTGGACTAAAATCTTGC R: TTTCACAGGTGCTGATAGGGA	60	[27]
TLR-2	F: ATCCTGCTGGAGCCCATTCAGAG R: TTGCTCTTCATCAGGAGGCCACTC	60	[28]
TLR-3	F: TCAGTACATTTGTAACACCCCGCC R: GGCGTCATAATCAAACACTCC	60	[28]
TLR-4	F: TGCCATCCCAACCCAACCACAG R: ACACCCACTGAGCAGCACCAA	60	[28]
TLR-21	F: CCTGCGCAAGTGTCCGCTCA R: GCCCCAGGTCCAGGAAGCAG	60	[28]

**Table 2 vetsci-12-00186-t002:** Heatmap depicting gene expression levels in the chicken intestinal tract across different developmental stages of chick embryos. Red indicates the lowest expression, green is the highest, and intermediate shades represent moderate expression levels. Blank cells indicate no detectable expression.

Gene	Embryonic Day	Duodenum	Jejunum	Ileum	Cecum
IFN-γ	ED14	4.11 × 10^−6^	1.98 × 10^−6^	1.51 × 10^−5^	4.08 × 10^−6^
	ED17	3.69 × 10^−6^	8.13 × 10^−6^	1.56 × 10^−6^	2.41 × 10^−6^
	ED20	1.68 × 10^−5^	1.98 × 10^−6^	2.05 × 10^−6^	7.79 × 10^−5^
IL-13	ED14	6.03 × 10^−7^	2.83 × 10^−7^	1.66 × 10^−6^	1.15 × 10^−6^
	ED17	2.55 × 10^−6^	1.34 × 10^−6^	1.38 × 10^−6^	1.30 × 10^−6^
	ED20	2.70 × 10^−6^	1.81 × 10^−6^	1.92 × 10^−6^	1.36 × 10^−5^
IL-6	ED14	1.22 × 10^−7^	1.75 × 10^−6^	3.55 × 10^−7^	1.43 × 10^−7^
	ED17	9.15 × 10^−8^	8.52 × 10^−8^	1.24 × 10^−7^	2.96 × 10^−7^
	ED20	8.10 × 10^−7^	1.19 × 10^−12^	3.55 × 10^−7^	2.25 × 10^−5^
TGF-β	ED14	6.34 × 10^−6^	9.08 × 10^−6^	6.55 × 10^−5^	1.43 × 10^−5^
	ED17	8.11 × 10^−6^	7.93 × 10^−5^	5.16 × 10^−6^	9.14 × 10^−6^
	ED20	1.87 × 10^−5^	4.63 × 10^−5^	4.62 × 10^−5^	1.30 × 10^−4^
IL-10	ED14		1.64 × 10^−8^	6.40 × 10^−8^	5.73 × 10^−8^
	ED17	2.54 × 10^−8^	4.88 × 10^−8^	7.37 × 10^−09^	2.74 × 10^−8^
	ED20		1.11 × 10^−6^	2.44 × 10^−7^	1.96 × 10^−6^
IL-8	ED14	8.60 × 10^−5^	1.12 × 10^−4^	4.44 × 10^−5^	3.94 × 10^−5^
	ED17	8.23 × 10^−5^	8.64 × 10^−6^	1.49 × 10^−5^	2.40 × 10^−5^
	ED20	1.81 × 10^−5^	2.22 × 10^−5^	5.05 × 10^−6^	6.36 × 10^−5^
AvBD-1	ED14	7.46 × 10^−3^	1.78 × 10^−3^	3.32 × 10^−3^	5.78 × 10^−3^
	ED17	2.80 × 10^−2^	9.18 × 10^−3^	5.25 × 10^−3^	1.34 × 10^−2^
	ED20	2.17 × 10^−5^	3.00 × 10^−2^	1.26 × 10^−2^	4.53 × 10^−5^
AvBD-2	ED14	2.42 × 10^−4^	6.06 × 10^−5^	7.40 × 10^−4^	2.72 × 10^−4^
	ED17	1.87 × 10^−3^	1.87 × 10^−4^	3.89 × 10^−4^	1.04 × 10^−3^
	ED20	2.17 × 10^−6^	2.00 × 10^−6^	1.20 × 10^−7^	5.00 × 10^−8^
AvBD-3	ED14	5.87 × 10^−6^	3.70 × 10^−6^	5.35 × 10^−5^	1.87 × 10^−5^
	ED17	2.82 × 10^−5^	4.86 × 10^−6^	3.95 × 10^−6^	1.83 × 10^−5^
	ED20	1.10 × 10^−6^	3.63 × 10^−5^	1.35 × 10^−5^	3.91 × 10^−6^
AvBD-4	ED14	8.96 × 10^−5^	2.60 × 10^−5^	4.81 × 10^−3^	6.54 × 10^−5^
	ED17	5.05 × 10^−4^	1.05 × 10^−4^	8.74 × 10^−5^	1.78 × 10^−4^
	ED20	1.87 × 10^−5^	6.82 × 10^−4^	2.56 × 10^−4^	2.77 × 10^−4^
AvBD-6	ED14	3.26 × 10^−5^	8.49 × 10^−6^	1.37 × 10^−1^	3.77 × 10^−5^
	ED17	2.73 × 10^−4^	1.51 × 10^−5^	1.29 × 10^−5^	7.16 × 10^−5^
	ED20	1.44 × 10^−5^	2.38 × 10^−3^	3.42 × 10^−5^	2.66 × 10^−5^
CATH-1	ED14	1.04 × 10^1^	7.52 × 10^−5^	2.23 × 10^−2^	2.96 × 10^−4^
	ED17	1.32 × 10^−3^	3.67 × 10^−4^	1.67 × 10^−4^	4.99 × 10^−4^
	ED20	8.56 × 10^−5^	2.95 × 10^−3^	6.58 × 10^−5^	1.47 × 10^−3^
CATH-2	ED14	3.97 × 10^−1^	5.92 × 10^−6^	7.37 × 10^−6^	2.10 × 10^−5^
	ED17	1.68 × 10^−4^	5.29 × 10^−7^	2.10 × 10^−5^	5.20 × 10^−5^
	ED20	1.22 × 10^−6^	4.39 × 10^−4^	2.96 × 10^−6^	9.71 × 10^−6^
CATH-3	ED14	6.24 × 10^−4^	1.89 × 10^3^	4.42 × 10^−3^	1.04 × 10^−3^
	ED17	4.11 × 10^−3^	1.70 × 10^−3^	5.47 × 10^−4^	1.85 × 10^−3^
	ED20	1.56 × 10^−4^	6.65 × 10^−3^	4.33 × 10^−4^	2.00 × 10^−3^
TLR-2	ED14	3.59 × 10^−4^	4.63 × 10^−4^	5.45 × 10^−4^	6.64 × 10^−4^
	ED17	6.12 × 10^−4^	3.21 × 10^−4^	5.24 × 10^−4^	6.30 × 10^−4^
	ED20	1.12 × 10^−4^	3.08 × 10^−4^	9.91 × 10^−8^	1.11 × 10^−4^
TLR-3	ED14	7.43 × 10^−5^	4.18 × 10^−4^	4.01 × 10^−4^	1.86 × 10^−3^
	ED17	1.52 × 10^−3^	9.95 × 10^−4^	5.91 × 10^−4^	1.86 × 10^−3^
	ED20	6.10 × 10^−4^	1.11 × 10^−3^	3.80 × 10^−3^	3.14 × 10^−3^
TLR-4	ED14	1.23 × 10^−4^	2.47 × 10^−4^	3.17 × 10^−4^	1.77 × 10^−4^
	ED17	2.27 × 10^−4^	5.52 × 10^−5^	9.52 × 10^−5^	1.51 × 10^−4^
	ED20	2.27 × 10^−4^	3.66 × 10^−5^	4.56 × 10^−5^	4.07 × 10^−3^
TLR-21	ED14	6.49 × 10^−6^	1.50 × 10^−5^	1.60 × 10^−5^	1.87 × 10^−5^
	ED17	6.99 × 10^−6^	5.29 × 10^−7^	6.07 × 10^−6^	5.03 × 10^−6^
	ED20	6.17 × 10^−7^	2.96 × 10^−7^	7.65 × 10^−10^	1.18 × 10^−6^

## Data Availability

Data available upon request.

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
