# Peer review of "Differential Expression of Key Immune Markers in the Intestinal Tract of Developing Chick Embryos"

_vetsci, 2025, doi:10.3390/vetsci12020186_

Round 1

Reviewer 1 Report

Comments and Suggestions for Authors

This study by Sharma et al. provides valuable insights into the developmental trajectory of intestinal immunity in chick embryos by evaluating basal gene expression levels of key immune markers, including cytokines, antimicrobial peptides (AMPs), and Toll-like receptors (TLRs), across embryonic days (ED) 14, 17, and 20. The findings reveal elevated cytokine expression in the cecum and increased AMP levels in the jejunum at ED20, offering a comprehensive perspective on the regional and temporal dynamics of immune gene expression.

The research addresses a critical gap in understanding innate immunity during embryonic development, presenting a solid foundation for exploring strategies to enhance immune function. However, several areas could be improved to enhance the clarity, impact, and relevance of the manuscript.

Specifically:

1.      Methodological Details: The methods used to evaluate basal gene expression levels should be further elaborated. For example, it is important to clarify whether quantitative PCR, RNA-seq, or other techniques were employed. The inclusion of Western blot (WB) analysis is also recommended to strengthen the findings and improve reproducibility.

2.      Discussion of Gene Variability: While the variability in the expression of AvBDs, CATHs, and TLRs across developmental stages is noted, further discussion is needed to explore potential regulatory mechanisms or environmental factors influencing this variability.

3.      Data Visualization: Including visual representations such as heatmaps or graphs to depict the expression patterns of cytokines, AMPs, and TLRs across intestinal regions and developmental stages would make the results more accessible and easier to interpret.

4.      Comparative Analysis: A comparison of the findings with data from other species or post-hatch chicks would contextualize the results and highlight unique aspects of intestinal immunity in chick embryos.

Overall, this study makes a significant contribution to avian immunology and provides a foundation for advancing immune system research in poultry. By addressing the identified concerns and recommendations, the manuscript could further enhance its scientific rigor and impact.

Author Response

This study by Sharma et al. provides valuable insights into the developmental trajectory of intestinal immunity in chick embryos by evaluating basal gene expression levels of key immune markers, including cytokines, antimicrobial peptides (AMPs), and Toll-like receptors (TLRs), across embryonic days (ED) 14, 17, and 20. The findings reveal elevated cytokine expression in the cecum and increased AMP levels in the jejunum at ED20, offering a comprehensive perspective on the regional and temporal dynamics of immune gene expression. The research addresses a critical gap in understanding innate immunity during embryonic development, presenting a solid foundation for exploring strategies to enhance immune function. However, several areas could be improved to enhance the clarity, impact, and relevance of the manuscript. Specifically: 1. Methodological Details: The methods used to evaluate basal gene expression levels should be further elaborated. For example, it is important to clarify whether quantitative PCR, RNA-seq, or other techniques were employed. The inclusion of Western blot (WB) analysis is also recommended to strengthen the findings and improve reproducibility. Thank you for your valuable feedback. We have clarified in line 121 that quantitative RT-PCR was used to evaluate gene expression in this study. While we acknowledge the suggestion to include Western blot (WB) analysis, the low levels of gene expression observed in our study may have rendered WB analysis unsuitable for reliably detecting the target proteins. This limitation is further compounded by the restricted availability of chicken-specific antibodies. We hope this explanation addresses the reviewer’s concerns and provides further clarity on our methodological approach. 2. Discussion of Gene Variability: While the variability in the expression of AvBDs, CATHs, and TLRs across developmental stages is noted, further discussion is needed to explore potential regulatory mechanisms or environmental factors influencing this variability. We acknowledge that this study did not include a detailed investigation of the regulatory mechanisms or environmental factors influencing this variability, as our primary focus was to characterize basal gene expression levels across key developmental time points to address the knowledge gap in immune gene expression dynamics during embryogenesis. Nonetheless, to the best of our knowledge, this is the first study to investigate intestinal immunity during embryogenesis, providing a foundational framework for future research utilizing transcriptomics, metabolomics, and proteomics to elucidate regulatory mechanisms and gain deeper insights into the factors driving variability in immune gene expression. 3. Data Visualization: Including visual representations such as heatmaps or graphs to depict the expression patterns of cytokines, AMPs, and TLRs across intestinal regions and developmental stages would make the results more accessible and easier to interpret. We appreciate the reviewer’s suggestion regarding data visualization. In the revised version, we have included a heatmap that depicts the expression patterns of cytokines across intestinal regions and developmental stages. We hope this approach aligns with the reviewer’s suggestion to ensure accessibility and clarity. 4. Comparative Analysis: A comparison of the findings with data from other species or post-hatch chicks would contextualize the results and highlight unique aspects of intestinal immunity in chick embryos. We thank the reviewer for their valuable input and would like to highlight that we have previously conducted this comparison in our published review article (reference provided below). Alkie, T. N., Yitbarek, A., Hodgins, D. C., Kulkarni, R. R., Taha-Abdelaziz, K., & Sharif, S. (2019). Development of innate immunity in chicken embryos and newly hatched chicks: A disease control perspective. Avian Pathology, 48(4), 288-310. https://doi.org/10.1080/03079457.2019.1607966 However, that review did not address the developmental trajectory of intestinal immunity of chick embryos. Our ongoing research aims to compare the expression levels of these genes both pre- and post-hatch, providing a more comprehensive understanding of their role in early immune development.

Reviewer 2 Report

Comments and Suggestions for Authors

It's an interesting study that focus on the developmental changes in the intestinal immune markes in chicken embryos. The topic falls into the scope of the journal. The manuscript is well prepared.

1. The measurements were conducted on ED 14, 17, and 20. The authors should provide the relevant evidance or reference.

2. Why was the measurement not perform in hatchlings ?

3. For the data analysis of different intestinal regions, the same eight embryos were compared. Hence, the repeated measurement analysis should be used. 

Author Response

It's an interesting study that focus on the developmental changes in the intestinal immune markes in chicken embryos. The topic falls into the scope of the journal. The manuscript is well prepared. 1. The measurements were conducted on ED 14, 17, and 20. The authors should provide the relevant evidence or references. Since no prior research has investigated intestinal immunity in embryos, this study serves as the first exploration in this field. Consequently, the absence of existing references made it challenging to determine the precise onset of gene expression, and it remains possible that these genes are expressed at earlier developmental stages. Despite this limitation, our findings provide a foundational framework for future studies to investigate earlier embryonic stages and further elucidate the ontogeny of intestinal immunity. We have acknowledged this limitation in the discussion section and emphasized the need for additional research to refine our understanding of the developmental timeline of immune gene expression in the intestine. We have acknowledged this limitation in the discussion section of our manuscript and emphasized the need for additional research to refine our understanding of intestinal ontogeny at earlier timepoints. 2. Why was the measurement not perform in hatchlings? The measurements were not performed in hatchlings because the primary focus of this study was on immunological development in the embryonic gut of chicks, which remains underexplored. Additionally, the immunological development in hatchlings has already been discussed in our previously published review article (Alkie, T. N., Yitbarek, A., Hodgins, D. C., Kulkarni, R. R., Taha-Abdelaziz, K., & Sharif, S. (2019). Development of innate immunity in chicken embryos and newly hatched chicks: A disease control perspective. Avian Pathology, 48(4), 288-310. ) https://doi.org/10.1080/03079457.2019.1607966 Since that review did not address the developmental trajectory of innate immune genes in the intestine of chick embryos, we aimed to address the gap during the embryonic stages. 3. For the data analysis of different intestinal regions, the same eight embryos were compared. Hence, the repeated measurement analysis should be used. We want to draw the reviewer's attention to the fact that we sampled different embryos at each time point. Repeated measures would have been applicable only if the same embryos were sampled/biopsied across all time points.

Reviewer 3 Report

Comments and Suggestions for Authors

Review‘s comments

The manuscript revealed the developmental trajectory of intestinal immunity in chick embryos by evaluating basal gene expression levels of key immune markers at different stages of development. Results indicated variable expression levels of cytokines, antimicrobial peptides (AMPs), and Toll-like receptors (TLRs) genes throughout the intestinal tract, which provides valuable insights into the basal level of various immune system gene expressions in the intestinal tract of chickens. These findings enhances our understanding of intestinal immunity during the embryonic stage and can inform the development of strategies to improve chick health by targeting early immune responses, filling this gap in innate immunity in the chicken gut during the embryonic period.

The work done in this study is significant and provides reference for the intestinal immunity during the embryonic stage. However, there are still some issues of concern that you need to explain.

1. lines 94-95: “Embryonated Ross 308 commercial broiler eggs were obtained from a hatchery (Fieldale Farms Corporation, GA)” There is an advantage in choosing SPF chicken embryos over commercial eggs to rule out the effects of vertical transmission of pathogens. As far as I know, many provenance diseases have a great influence on chicken intestinal immunity.

2. Lines 99-100: Is there any basis or reason for choosing the 14,17 and 20 days of age of embryonic development.

3. Lines 227-232: The gene expression of another toll-like receptor, TLR3 (recognizes dsRNA), should be further examined, which is also critical for evaluating the intestinal immunity of chicken embryos.

4. Lines 460-461, 494-496, 520-522: Please fill in the missing page numbers of this references.

Author Response

The manuscript revealed the developmental trajectory of intestinal immunity in chick embryos by evaluating basal gene expression levels of key immune markers at different stages of development. Results indicated variable expression levels of cytokines, antimicrobial peptides (AMPs), and Toll-like receptors (TLRs) genes throughout the intestinal tract, which provides valuable insights into the basal level of various immune system gene expressions in the intestinal tract of chickens. These findings enhances our understanding of intestinal immunity during the embryonic stage and can inform the development of strategies to improve chick health by targeting early immune responses, filling this gap in innate immunity in the chicken gut during the embryonic period. The work done in this study is significant and provides reference for the intestinal immunity during the embryonic stage. However, there are still some issues of concern that you need to explain. 1. lines 94-95: “Embryonated Ross 308 commercial broiler eggs were obtained from a hatchery (Fieldale Farms Corporation, GA)” There is an advantage in choosing SPF chicken embryos over commercial eggs to rule out the effects of vertical transmission of pathogens. As far as I know, many provenance diseases have a great influence on chicken intestinal immunity. Thank you for your insightful comment. We intentionally chose commercial broiler eggs over SPF eggs to better simulate conditions encountered in commercial poultry production. While we acknowledge that SPF embryos would eliminate concerns regarding vertical transmission of pathogens, our aim was to investigate intestinal immunity under conditions representative of the commercial industry. This approach allows us to address practical challenges faced in real-world settings and ensures the broader applicability of our findings. 2. Lines 99-100: Is there any basis or reason for choosing the 14,17 and 20 days of age of embryonic development. Since no prior research has investigated intestinal immunity in embryos, this study serves as the first study to explore intestinal immunity during embryogenesis. As a result, we could not determine the precise onset of gene expression, and it is possible that these genes are expressed at earlier developmental stages. Despite this limitation, our findings provide a foundational framework for future studies to investigate earlier embryonic stages and further elucidate the ontogeny of intestinal immunity. We have acknowledged this limitation in the discussion section and emphasized the need for additional research to refine our understanding of the developmental timeline of immune gene expression in the intestine. 3. Lines 227-232: The gene expression of another toll-like receptor, TLR3 (recognizes dsRNA), should be further examined, which is also critical for evaluating the intestinal immunity of chicken embryos. As requested by the reviewer, we have measured TLR3 and incorporated the findings into the revised manuscript.

Round 2

Reviewer 1 Report

Comments and Suggestions for Authors

The authors have addressed the questions effectively and made appropriate revisions to the manuscript. Based on this, I believe there are no further issues and I recommend accepting and publishing the manuscript.

Reviewer 2 Report

Comments and Suggestions for Authors

No further comment.